# Regulation of Primary Cilium Length by O-GlcNAc during Neuronal Development in a Human Neuron Model

**DOI:** 10.3390/cells12111520

**Published:** 2023-05-31

**Authors:** Jie L. Tian, Chia-Wei Huang, Farzad Eslami, Michael Philip Mannino, Rebecca Lee Mai, Gerald W. Hart

**Affiliations:** 1Complex Carbohydrate Research Center, University of Georgia, Athens, GA 30602, USA; chiawei.huang@uga.edu (C.-W.H.); farzadig@uga.edu (F.E.); michael.mannino@uga.edu (M.P.M.); rebecca.mai@uga.edu (R.L.M.); 2Department of Biochemistry and Molecular Biology, University of Georgia, Athens, GA 30602, USA; 3Department of Biology, University of Georgia, Athens, GA 30602, USA

**Keywords:** O-GlcNAc, primary cilia, neuronal development, cortical neurons, human induced-pluripotent stem cells

## Abstract

The primary cilium plays critical roles in the homeostasis and development of neurons. Recent studies demonstrate that cilium length is regulated by the metabolic state of cells, as dictated by processes such as glucose flux and O-GlcNAcylation (OGN). The study of cilium length regulation during neuron development, however, has been an area left largely unexplored. This project aims to elucidate the roles of O-GlcNAc in neuronal development through its regulation of the primary cilium. Here, we present findings suggesting that OGN levels negatively regulate cilium length on differentiated cortical neurons derived from human-induced pluripotent stem cells. In neurons, cilium length increased significantly during maturation (after day 35), while OGN levels began to drop. Long-term perturbation of OGN via drugs, which inhibit or promote its cycling, during neuron development also have varying effects. Diminishing OGN levels increases cilium length until day 25, when neural stem cells expand and undergo early neurogenesis, before causing cell cycle exit defects and multinucleation. Elevating OGN levels induces greater primary cilia assembly but ultimately results in the development of premature neurons, which have higher insulin sensitivity. These results indicate that OGN levels and primary cilium length are jointly critical in proper neuron development and function. Understanding the interplays between these two nutrient sensors, O-GlcNAc and the primary cilium, during neuron development is important in paving connections between dysfunctional nutrient-sensing and early neurological disorders.

## 1. Introduction

The primary cilium, a solitary and non-mobile appendage, protrudes from the surface of nearly all mammalian cells [1]. Though overlooked as vestigial remnants of a long-lost and motile counterpart for decades following their discovery, phenotypes of cilia mutants have encouraged modern researchers to grasp a greater comprehension of ciliogenesis and the influence of developed primary cilia on physiological processes. Besides timing of cilia assembly and disassembly, which largely hinges on a bidirectional crosstalk with cell division, structural changes in cilia also bear tremendous implications in the overall function of the organelle [2,3,4,5,6,7]. To properly fulfill their various roles, primary cilia reach a specified range of sizes depending on the cell type from which they extend [8,9,10]. Disruptions in the form or function of these projections commonly result in a myriad of debilitating disorders based on the affected organ, including cognitive deficiencies, blindness, deafness, and irregular breathing patterns, termed ciliopathies [11,12,13,14,15]. Beyond their transductory roles in special sensory cells, enabling the sensation of movement, light, odors, and various additional stimuli, primary cilia are instrumental in regulating several neurodevelopmental pathways [16,17,18,19,20,21,22]. Ablation and reduction of primary cilia through conditional knockout of certain genes such as Stumpy, which encodes a basal body protein B9 domain-containing protein 2, diminish hippocampal neurogenesis through an impairment of Hedgehog signaling [23,24]. In the nervous system, cilia are essential for proper patterning, maturation, general survival, and stem cell fate determination, as explained by their links with non-canonical Wnt, Hippo, Notch, mTOR, and TGF-β, among others [25,26,27,28]. Primary cilia are also highly dynamic and undergo alterations in shape, size, and composition constantly [29,30]. Such changes are brought about by enzymatic cascades, gene modifications, and even post-translational modifications on the cilium itself [31,32,33,34,35].

OGN, or the incorporation of O-linked-β-N-acetylglucosamine (O-GlcNAc) onto serine and threonine residues, is a post-translational modification which modulates protein function [36,37,38,39]. Despite being observed on over 9000 human proteins since its finding, the inclusion and cleavage of this moiety are driven by only a pair of enzymes: O-GlcNAc transferase (OGT) and O-GlcNAcase (OGA), respectively [40]. OGT’s sensitivity to Uridine diphosphate N-acetylglucosamine (UDP-GlcNAc), its donor substrate and a central node in metabolism connecting glucose, fatty acid, amino acid, and nucleotide productions, makes OGN a ‘rheostat’ capable of adjusting cellular activity when faced with stress and/or changes in nutrient availability. In instances of glucose starvation, for example, O-GlcNAc modification of most proteins also tapers down, while a surplus of glucose results in enhanced OGN [41]. Prolonged dysregulation of OGN has been observed in diabetes, Alzheimer’s disease, cardiomyopathy, virtually all cancers, and other chronic illnesses of aging [42,43,44,45,46,47,48,49]. This single sugar addition regulates a diverse set of essential biological processes including transcription, mitochondrial functions, cell cycle progression, and cilia assembly [36,50,51,52,53,54]. Several studies have explored the separate effects of aberrant OGN on primary cilia and neuronal developments, but none have integrated the two. The following works attempt to record and understand how nutrient flux and inhibition of OGT/OGA impact the differentiation of pluripotent stem cells into mature neurons. Investigating how altered O-GlcNAc cycling affects primary cilia formation and subsequent neural development holds potential in uncovering clues in how deficient metabolism gives rise to disease.

## 2. Materials and Methods

### 2.1. iPSC Cell Lines and Maintenance

Human iPSC line K3 (Sex: male), a kind gift from Dr. Michael Tiemeyer at the University of Georgia, was previously established and characterized [55]. The cells were maintained as previously described [56]. Briefly, cells were cultured in feeder-free and chemically defined Essential 8 medium (A15710-01, Gibco, Grand Island, NY, USA) on 5 μg/mL vitronectin coated (A14700, Gibco, Grand Island, NY, USA) cell culture plates and were passaged with 0.5 mM EDTA (46-034-Cl, Corning, Corning, NY, USA). The cells were maintained with 5% CO_2_ in a 37 °C incubator.

### 2.2. Cortical Neuron Differentiation and Treatments

Human iPSCs were induced into cortical neurons according to established protocols [57,58]. Briefly, when the cells reached 70–90% confluency, cells on two dishes were replated onto a single Geltrex (1:200, A1413302, Gibco, Grand Island, NY, USA)-coated culture plates. One-hundred percent confluency was evaluated after 24 h replating. After washing the cells with DPBS twice, a neural induction medium consisting of neural maintenance medium, 10 μM SB431542 (16-141-0, Tocris, Minneapolis, MN, USA), and 1 μM Dorsomorphin (30-931-0, Tocris, Minneapolis, MN, USA) was added to the cells. The neural maintenance medium contained 1:1 mixture of DMEM/F-12 GlutaMAX (10565018, Gibco, Grand Island, NY, USA) and Neurobasal medium (12348-017, Gibco, Grand Island, NY, USA), 0.5× B-27 (17504044, Gibco, Grand Island, NY, USA), 0.5× N-2 (17502048, Gibco, Grand Island, NY, USA), 0.5× Non-essential amino acids (11140-050, Gibco, Grand Island, NY, USA), 0.5× GlutaMAX (35050-061, Gibco, Grand Island, NY, USA), 500 mM sodium pyruvate (S8636, Sigma-Aldrich, Burlington, MA, USA), 2.5 mg/mL Insulin (I9278, Sigma-Aldrich, Burlington, MA, USA), and 50 μM 2-mercaptoethanol (21985023, Gibco, Grand Island, NY, USA), P/S (25 U/mL, 15140122, Gibco, Grand Island, NY, USA). On day 12, cells were detached using dispase (7913, STEMCELL Technologies, Vancouver, BC, Canada) and replated with neural induction medium onto laminin-coated culture plates (10 mg/mL, L2020, Sigma-Aldrich, Burlington, MA, USA). 20 ng/mL of basic-FGF (4114-TC, R&D Systems, Minneapolis, MN, USA) was added to neural maintenance medium from day 13 to 16. On day 17, basic-FGF was withdrawn from the medium, and the cells were maintained in only neural maintenance medium. On day 25, cells were passaged as single cells with Accutase (25-058-CI, Corning, Corning, NY, USA) onto laminin-coated culture plates. On day 35, the cells were replated at a density of 100 K cells/cm^2^ onto poly-L-ornithine (P3655, Sigma-Aldrich, Burlington, MA, USA) and laminin-coated culture plates. For the following days until day 55 (the end of differentiation), cells were cultured in neural maintenance medium. For short-term OGN alteration, Ac-5SGlcNAc (Ac5S, 50 μM, a kind gift from Dr. Boons) or Thiamet-G (TMG, 1 μM, SML0244, Sigma-Aldrich, Burlington, MA, USA) were used to treat neurons for 24 h on day 55 after differentiation. For long-term perturbation of O-GlcNAc levels, Ac5S (25 μM) or TMG (1 μM) were added to the culture medium from day 0 until the end of the differentiation process. For insulin treatment, cells were starved for 3 h followed by insulin (I9278, Sigma-Aldrich, Burlington, MA, USA) treatment at the indicated concentration for 30 min on day 55 of differentiation. 

### 2.3. Western Blots

Cells were lysed in RIPA buffer with 1× protease inhibitor (A32963, Thermo Scientific, Waltham, MA, USA), 1× phosphatase inhibitor (A32957, Thermo Scientific; Waltham, MA, USA), and 10 μM PUGNAc (A7229, Sigma-Aldrich, Burlington, MA, USA) and then centrifuged at 13,200 rpm for 15 min at 4 °C. The supernatant was collected, and total protein concentration was measured using a BCA Protein Assay kit (23227, Thermo Scientific; Waltham, MA, USA), according to the manufacturer’s protocols. For Western blot experiments, 20 μg of total cell protein was combined with 4× Laemmli Sample Buffer (Bio-Rad, 1610747), diluted to 1× concentration, and heated at 95 °C for 10 min. The samples were separated on 4–12% pre-cast Bis-Tris gels (WG1402BOX, Invitrogen, Carlsbad, CA, USA) at 120V for ~90 min, transferred onto 0.2 mm PVDF membranes (10-6000-21, GE healthcare, Chicago, IL, USA) at 400 mA for 2 h, and blocked with 5% BSA (BSA1000, Rockland Immunochemicals, Pottstown, PA, USA). The membranes were incubated with primary antibodies overnight at 4 °C. They were washed three times with TBS-T (10 min each) and incubated with HRP secondary antibodies for 1 h at room temperature. After three washes with TBS-T (10 min each), the samples were imaged. Antibodies used for Western blotting analysis were specific to Nestin (1:1000, MO22183, Neuromics, Edina, MN, USA), Tuj1 (1:1000, 802001, BioLegend, San Diego, CA, USA), CTIP2 (1:1000, 650602, BioLegend, San Diego, CA, USA), SatB2 (1:500, sc-81376, Santa Cruz, Dallas, TX, USA), Caspase3 (1: 1000, 9662S, Cell Signaling, Danvers, MA, USA), O-GlcNAc (1:1000, CTD110.6, made in house), and β-actin (1:5000, 3700S, Cell Signaling, Danvers, MA, USA). Membrane imaging was completed using the iBright FL1500 imaging system (Invitrogen, Carlsbad, CA, USA). Signal detection was developed using SuperSignal West Pico PLUS Chemiluminescent Substrate (34578, Thermo Scientific, Waltham, MA, USA), and band intensity was quantified with NIH ImageJ (Version 2.3.0, a public domain image analysis software) and normalized to β-actin.

### 2.4. Immunofluorescence Staining

Immunofluorescence staining was carried out according to previous study [59]. Cell samples were fixed in methanol (34860, Sigma-Aldrich, Burlington, MA, USA) for 10 min, permeabilized with 0.25% Triton X-100 (T9284, Sigma-Aldrich, Burlington, MA, USA) in PBS, washed with PBS, and blocked for 30 min in blocking buffer containing 2% BSA (BSA1000, Rockland Immunochemicals, Pottstown, PA, USA), 0.2% gelatin (G1890, Sigma-Aldrich, Burlington, MA, USA), 10 mM glycine (G8898, Sigma-Aldrich, Burlington, MA, USA), and 50 mM NH_4_Cl (393182500, Acros Organics, Geel, Belgium). The samples were incubated with primary antibodies in blocking buffer overnight at 4 °C. After the samples were washed three times, they were incubated with fluorescent secondary antibodies and DAPI staining solution at room temperature. Antibodies used for immunofluorescence staining analysis were specific to NeuN (1:500, ab104224, Abcam, Cambridge, UK), Tuj1 (1:500, 802001, BioLegend, San Diego, CA, USA), Arl13b (1:250, 66739-1-Ig, Proteintech, Rosemont, IL, USA), and IFT88 (1:200, 13967-1-AP, Proteintech, Rosemont, IL, USA). Stained cells were post-fixed with 2% PFA in PBS for 10 min, followed by imaging with an Echo Revolve microscope (San Diego, CA, USA), and images were analyzed with ImageJ (Version 2.3.0) to measure cilium length and multinucleated cells percentage. 

### 2.5. Quantification and Statistical Analysis

The GraphPad Prism software (Version 9.5.1) was used for statistical analysis. All statistical analyses were performed using *t*-test, one-way, or two-way ANOVA followed by multiple comparison test. Data in graphs are expressed as mean values ± standard deviation of the mean (mean ± SD). Error bars represented denote the SD. 

## 3. Results

### 3.1. Results

#### 3.1.1. Cilium Length in Cortical Neurons Is Negatively Correlated with O-GlcNAc Levels

To study neuronal primary cilia, we differentiated human induced pluripotent stem cells (iPSC) into cortical neurons before culturing them in 30 mM (high glucose level), 5 mM (normal physiological glucose level in the brain), and 2 mM (low glucose level) glucose medium for 24 h to induce ciliogenesis. The formation of cilia on neurons was analyzed by immunostaining with antibodies against IFT88 and Arl13b cilia markers as well as neuronal nuclear protein (NeuN), a neuronal marker (Figure 1A). We determined primary cilium length from NeuN-positive cells by measuring the distance from the base to the tip as indicated by IFT88, which excels at staining the entire cilia with the disadvantage of having a high background that makes it difficult to tell cilia apart. For this reason, we also used Arl13, which stains robust cilia axonemes, as a complement in order to locate and measure cilium length more specifically and accurately. Cells cultured in 30 mM glucose had a primary cilium length of 2.11 ± 0.82 μm, while cells cultured in 5 mM and 2 mM glucose had significantly longer cilium lengths of 2.31 ± 0.90 μm and 2.60 ± 0.95 μm, respectively. Since glucose deprivation decreases hexosamine biosynthesis pathway (HBP) flux, dampening the O-GlcNAc modification of proteins, we detected cellular OGN levels at different glucose concentrations via Western blotting. The immunoblots suggested that cellular OGN levels decreased significantly when cells were cultured in 5 mM and 2 mM glucose medium when compared to 30 mM glucose (Figure 1B). To verify that elongation of neuronal cilium length was indeed caused by changes in O-GlcNAc cycling, we treated cells with OGT and OGA inhibitors, Ac5S and TMG, to decrease or increase OGN levels, respectively. After confirming the efficiency of chemical inhibitors by immunoblotting against O-GlcNAc (Figure 1C), we analyzed the neuronal cilium length (Figure 1D). The mean length of the 5 mM control group was 2.26 ± 0.87 μm. The Ac5S treatment group had an elongated cilium length of 2.61 ± 0.85 μm, while the TMG treatment group had a shortened cilium length of 1.64 ± 0.77 μm. Both differences are statistically significant. These results indicate that primary cilium length on cortical neuron cells is negatively regulated by cellular OGN levels.

#### 3.1.2. Cilia Formation Is Partially Regulated by O-GlcNAc Levels during Cortical Neuron Differentiation

After demonstrating that short-term alterations in OGN levels negatively regulate cilium length in differentiated cortical neurons, we wondered if OGN levels also display other trends inversely correlated with cilium length during cortical neuron differentiation. To answer this question, we utilized human iPSC-derived cortical neurons as a platform to observe the interaction between cilia and OGN during cortical neuron differentiation. Throughout the differentiation process, cells underwent several stages, including neural stem cells, progenitor cells, early phase neurons, and functional neurons (Figure 2A). Cells were harvested at several key time points to determine the OGN levels via Western blotting (Figure 2B). The results showed that O-GlcNAc modification levels gradually increased after differentiation and reached their highest levels at around day 35. After that, O-GlcNAc levels started to descend as the neuron matured. Conversely, cilium length was elongated from 1.85 ± 0.61 on day 0 to 2.12 ± 0.54 µm on day 12 and then gradually shortened to 1.85 ± 0.75 µm on day 35 at progenitor/early phase neuronal stages, while longer cilia were observed as the neurons matured over time (2.33 ± 0.90 µm on day 45 and 2.27 ± 0.79 µm on day 55) (Figure 2C). These results indicate that OGN levels might also be negatively associated with cilium length during neuron differentiation and, thereby, mediate multiple signaling pathways in the complex process of neuronal development.

#### 3.1.3. Long-Term Perturbation of OGN Causes Ciliary Defect during Cortical Neuron Development

To further investigate the relationship between OGN and cilia of cortical neurons during development, we turned to pharmacological means of regulating OGN levels. The cortical neurons were generated from iPSCs using the same protocol but with the addition of an OGT inhibitor, Ac5S, or OGA inhibitor, TMG, throughout the differentiation process (Figure 3A). The efficacy of inhibitors was confirmed by Western blotting at multiple key points of differentiation (Figure 3B). OGN levels were effectively diminished by Ac5S treatment and elevated by TMG administration at all key time points. The lengths of cilia on cells which had differentiated, either in the presence of Ac5S or TMG, were measured at several time points to reveal the effects of long-term OGN alteration (Figure 3C). On day 0 and day 12, Ac5S treatment groups displayed significantly elongated cilium length (2.11 ± 0.77 µm on day 0 and 2.27 ± 0.49 µm on day 12) when compared to control groups (1.89± 0.64 µm on day 0 and 2.05 ± 0.48 µm on day 12), while TMG treatments shortened cilium length (1.51 ± 0.49 µm and 1.51 ± 0.40 µm, respectively). On day 25, mean cilium lengths of both Ac5S-treated and TMG-treated cells were remarkably longer (2.21 ± 0.60 µm for Ac5S and 2.19 ± 0.89 µm for TMG) than control cells (1.98 ± 0.61 µm). Interestingly, we observed a dramatically increased percentage of multinucleated cells at 69.27%, 74.47%, and 80.83% by Ac5S treatment on days 35, 45, and 55, respectively. OGN and the cell cycle are inextricably linked, as a multitude of cell cycle-related proteins are dynamically modified by O-GlcNAc [60]. Hence, it is possible that abolishing O-GlcNAc directly affects cell division in early phases of differentiation. Since cilia were absent on the multinucleated cells of Ac5S-treated groups, we only compared cilia on control and TMG-treated cells after day 35. The TMG-treated cells displayed longer cilia, with mean lengths of 2.28 ± 0.92 µm on day 35, 2.79 ± 1.08 µm on day 45, and 3.37 ± 1.10 µm on day 55. Meanwhile, control cells had cilia measuring 1.85 ± 0.76 µm on day 35, 2.32 ± 0.90 µm on day 45, and 2.25 ± 0.79 µm on day 55. Altogether, decreased OGN caused aberrant cell cycle regulation and diminished cilia formation, while raised O-GlcNAc levels by TMG resulted in the formation of elongated cilia during cortical development.

#### 3.1.4. Altered OGN and Subsequent Cilia Formation Interferes with Cortical Neuronal Differentiation

To observe the impact of OGN and cilia formation on cortical neurogenesis, we harvested cells differentiated with and without Ac5S or TMG treatments on day 55. We tracked neuronal generation by immunostaining against Tuj1, a neuronal marker. Both Tuj1 staining and phase images indicated that the generation of neurons in the control group was robust. Ac5S-treated cells barely differentiated into neurons, with most of the cells developing into multinucleated fibroblast-like cells instead (Figure 4A). Upon further analysis by Western blotting, we found that neurons generated with TMG expressed neural progenitor marker, Nestin, and Tuj1 but lacked cortical neuronal markers CTIP2 and SatB2 (Figure 4B). Additionally, a higher level of cleaved-caspase3 was observed in TMG-treated neurons than in control neurons (Figure 4C). These results suggest that longer primary cilia, as induced by TMG, might result in premature differentiation and a reduced pool of progenitors, before ultimately triggering an abnormal fate specification and apoptosis. These results are congruent with findings in patients with Seckel syndrome, in which the mutation of a centrosomal-P4.1-associated protein (CPAP) hampers timely cilia disassembly [61]. In short, downregulating OGN during cortical neurogenesis may block cell cycle progression and result in multinucleated cells instead of neurons, while excessive OGN and, therefore, longer cilia lead to premature neurons with poor specification (Figure 4D). Our results emphasize the importance of finely coordinated OGN as well as cilia assembly and disassembly in the proper development of the human cortical neuron.

#### 3.1.5. Increased Primary Cilium Length Leads to Higher Insulin Sensitivity of TMG-Treated Neurons

Cilia are known to harbor many signal cascade-initiating receptors such as Wnt, TGF-β, IR, IGF-R1, and more [62]. Consequently, various signaling pathways have been shown to diminish when cilia are knocked out [63]. Knowing this, we wanted to determine if TMG-differentiated cells would show altered signaling due to their extended cilia. To do this, cells were serum starved and stimulated with insulin resulting in excitation of the PI3K/Akt pathway. Although both control and TMG-treated cells exhibited an increase in pT308 Akt phosphorylation upon insulin stimulation, the increase was 2.5-fold higher for TMG-treatment groups compared to a 1.4-fold increase for the control (Figure 5). This result may suggest that the difference in Akt signaling was due to the extended cilia.

## 4. Discussion

In this study, we demonstrated that primary cilium assembly on differentiated cortical neurons was negatively regulated through either glucose or OGN levels and that the length was partially correlated with OGN levels during neuron differentiation. Furthermore, differentiating neurons when OGN levels are low caused cell cycle alterations and reduced cilia formation; higher OGN levels induced premature neurons and elongated cilia, which are more sensitive to insulin stimulation. Altogether, these findings suggest that regulations of cilium length have different mechanisms under short-term and long-term disruptions in OGN.

This series of experiments shows that glucose and OGN levels have negative impacts on cortical neuronal cilium length, which is consistent with previous studies stating that higher glucose and OGN levels significantly decrease primary cilium length in human retinal pigment epithelial cells, diabetic mice, and diabetic patients [33,54]. Primary cilia of neurons are critical for regulating energy balance by sensing hormones and nutritional levels through various receptors, including the insulin receptor (IR) and leptin receptor (LepR) [64,65]. Dysfunction in neuronal cilia results in metabolic disorder-related obesity and diabetes due to the mislocalization of hormone receptors and compromised satiety response [66]. Combined with the study of elevated OGN levels dampening the insulin signaling pathway, we can hypothesize that O-GlcNAc is a negative regulator of primary cilium length on neurons. It has been previously shown that chow-diet-fed lean mice have longer cilium length while obese mice, with a high fat and sucrose diet, have shorter cilium length on hypothalamus cells and are leptin resistant [67]. Future work focusing on studying the relationship between the conditional knockout of OGT and cilium length in neurons will aid us in better understanding the mechanisms of ciliopathy-associated obesity and diabetes.

In addition to their importance on differentiated neurons, primary cilia also play crucial roles during brain development through the transduction of important signaling pathways, such as Wnt and Shh signaling [68]. Primary cilia are required for the differentiation of neural progenitors and neural stem cells into multiple types of brain cells [69,70]. Dysfunction of primary cilia in early brain development induces a plethora of severe abnormalities, such as neuronal tube defects, corpus callosum agenesis, cerebellar hypoplasia, and hydrocephalus [71]. Perhaps due to the diverse ways in which ciliary defects present themselves, very little is known about how cilium length is regulated during neuron differentiation. Here, we used a human iPSC model and showed that cilium length was indeed oscillating during the neuron differentiation process, rather than following a continuous trend. Cilia elongated significantly on days 45 and 55 in differentiated neurons, while OGN levels fell significantly compared to that of cells still undergoing differentiation, at day 35. These results suggest that O-GlcNAc is one of the factors involved in regulating cilium length during neuron differentiation. Future studies could investigate the molecular mechanisms behind how OGN levels regulate cilium length and whether primary cilia transduce different signals at different stages of differentiation. Interestingly, the primary cilium may also facilitate cell–cell communication functions in addition to its role as signal hub to transduce signals [68,72]. Our immunostaining results showed that neuronal cilia form clusters (Figure 2C, Day 55), which possibly allows for cell–cell communication.

Differentiating neurons with abnormal OGN levels for extended periods of time showed that cilium length and cell morphology were drastically different from those of short-term conditions. Differentiating neurons in conditions limiting OGN lead to longer cilium length until day 25, and the gradual death of cells; surviving cells were mostly de-ciliated and multinucleated. It has been previously known that attenuated cellular OGN levels result in multinucleation by blocking cytokinesis [73]. This could be explained by the prolonged depletion of OGN arresting the cell cycle by inhibiting cytokinesis and, thus, prohibiting neuron differentiation. On the other hand, when differentiating neurons underwent prolonged elevations in OGN levels, we observed that cells had significantly longer cilium length compared to the control group from day 25 and on. Cells also developed into neuron progenitors earlier but were unable to develop into cortical neurons. This is consistent with previous findings that long-cilia-iPSCs derived from Seckel syndrome patients showed signs of early differentiation, decreased proliferative capacity, and delayed cell cycle re-entry [61]. These results suggest that long-term TMG treatment has different effects on primary cilium length regulation compared to short-term treatment. Future research on how fluctuations in OGN levels regulate cilium length through cell cycle progression holds potential in greatly expanding our knowledge of neuron differentiation and its relationship with disease.

Even though our paper showed a negative correlation between OGN levels and primary cilium length during both mature and differentiating neurons, the molecular mechanisms behind how OGN’s regulation of cilium length remain a mystery. Future in vivo studies may investigate whether OGN regulates cilium length through the axoneme or by affecting receptor distribution on the primary cilia membrane. We are also interested in performing a cause-and-effect study on neurons differentiating with long-term aberrations in OGN levels, as it would be interesting to see whether cell multinucleation is caused by decreased OGN levels or by long primary cilia. Our research primarily focused on differentiating human iPSCs into cortical neurons as an in vitro model system to study the regulation of cilium length on human neurons by OGN levels and its potential effects. While iPSC-derived neurons have been extensively utilized as a valuable tool for investigating human neuronal development and disease mechanisms, it is important to acknowledge the inherent limitations of in vitro models when attempting to extrapolate findings to the complexity of in vivo systems.

## Figures and Tables

**Figure 1 cells-12-01520-f001:**
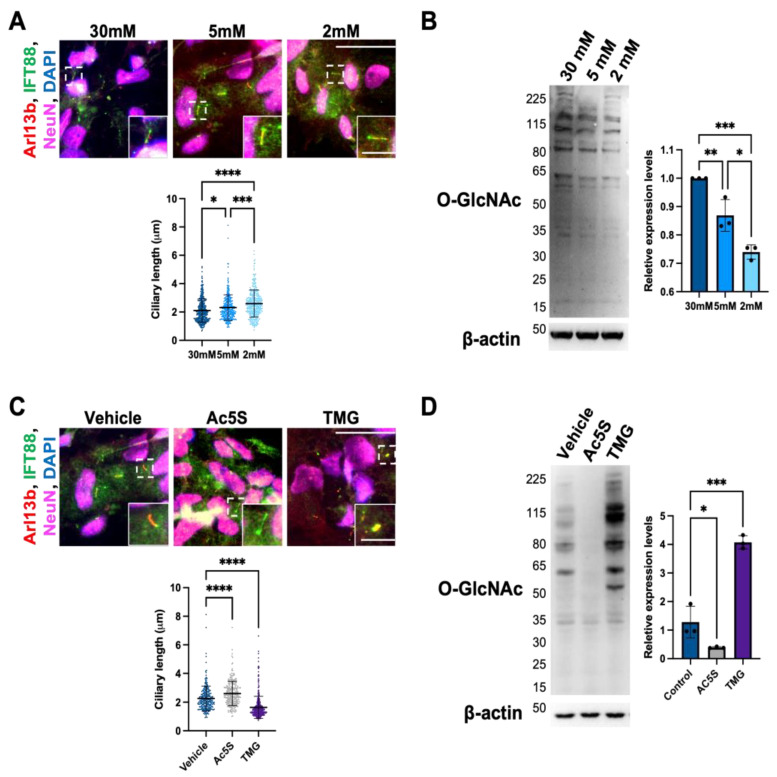
Cilium length in cortical neurons is negatively correlated with O-GlcNAc levels. (**A**) Immunofluorescent images of primary cilia on cortical neurons at different glucose concentrations; cells were co-immunostained with antibodies against NeuN, Arl13, and IFT88. Scatter plots represent the quantification of mean cilium lengths. (**B**) Glucose deprivation decreased cellular OGN levels. Cortical neurons were cultured in the medium with a glucose concentration of 30 mM, 5 mM, or 2 mM for 24 h. Cell lysates were analyzed by Western blotting with O-GlcNAc antibody; β-actin was used as loading control. Bar graph shows that the statistical analysis of O-GlcNAc level differences is significant. (**C**) OGN levels negatively regulate neuronal cilium length. Immunofluorescent image of cortical neuron primary cilia at different OGN levels. Scatter plots represent the quantification of mean cilium lengths. (**D**) Effects of OGT and OGA chemical inhibitors on cellular OGN levels. Cortical neurons were treated with and without inhibitors, Ac5S or TMG, for 24 h. Cell lysates were analyzed by Western blotting with O-GlcNAc antibody; β-actin was used as loading control. Bar graphs show that the statistical analysis of O-GlcNAc level differences is significant. Scale bars for the image are 40 µm and for the magnified images are 10 µm. All data are mean ± SD from three independent experiments (>80 cells per experiment). * *p* < 0.05; ** *p* < 0.01; *** *p* < 0.001; **** *p* < 0.0001; n.s., not significant.

**Figure 2 cells-12-01520-f002:**
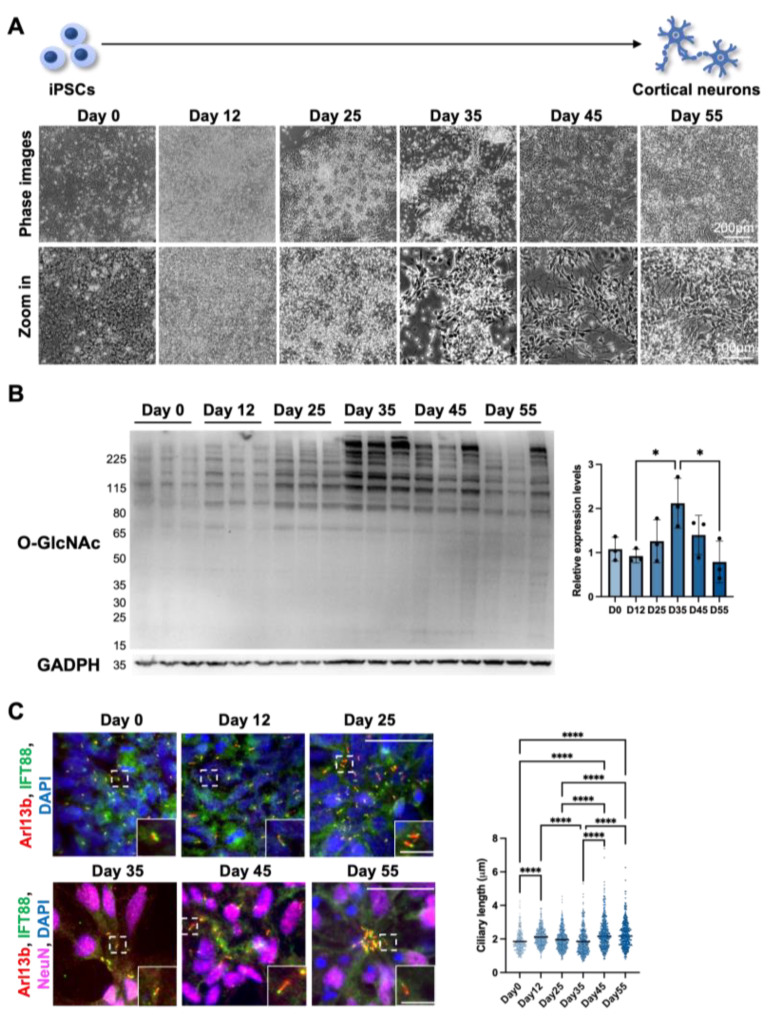
Cilia formation is partially regulated by O-GlcNAc levels during cortical neuron differentiation. (**A**) Schematics and phase images of iPSC-derived cortical neurons at key time points. Day 0: undifferentiated iPSC; Day 12: neuroepithelial cells; Day 25: rosette, which is composed of cortical stem/progenitor cells; Day 35–55: cortical neurons differentiation and maturation over time. (**B**) Western blotting result and quantification. Cells were harvested at indicated time points during differentiation, and O-GlcNAc levels were analyzed via Western blotting. (**C**) Representative images of immunofluorescence staining and the quantification of cilium length. Cells were harvested on days 0, 12, 25, 34, 45, and 55 and stained for IFT88, Arl13b, and NeuN (after day 25). Cilium lengths were measured on every ciliated cell on days 0, 12, and 25. Cilium lengths were measured on NeuN-positive cells after day 25. Scale bars for the image are 40 µm and for the magnified images are 10 µm. All data are mean ± SD from three independent experiments (>80 cells per experiment). * *p* < 0.05; **** *p* < 0.0001; n.s., not significant.

**Figure 3 cells-12-01520-f003:**
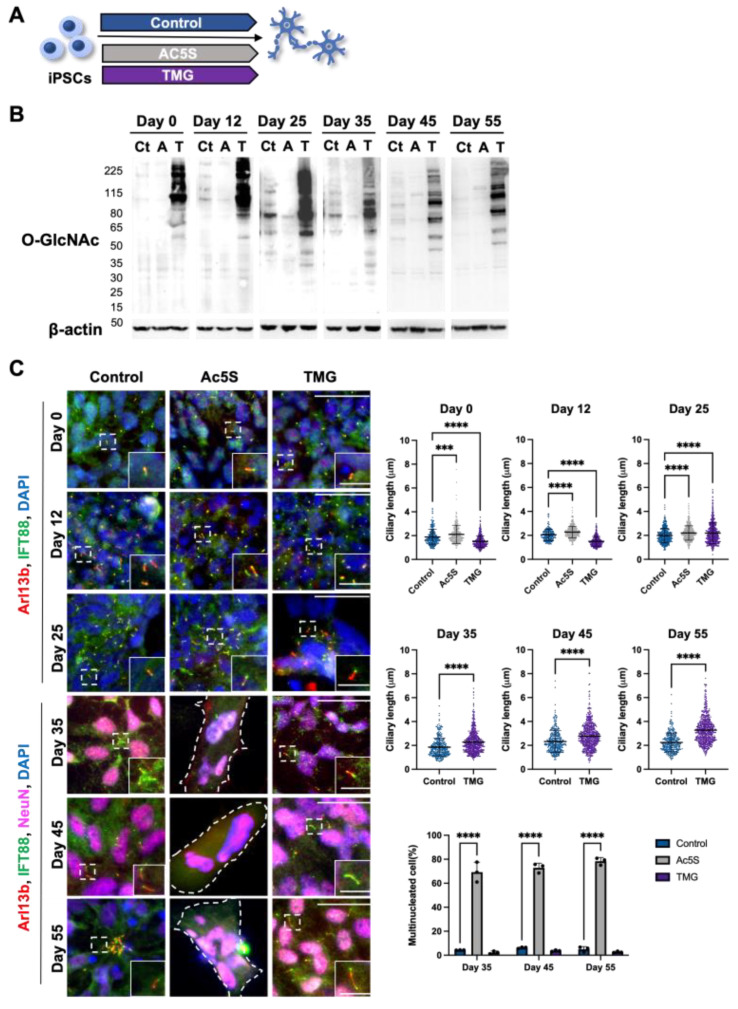
Long-term perturbation of OGN causes ciliary defect during cortical neuron development. (**A**) Schematic illustrating Ac5S and TMG treatment during cortical neuron differentiation. (**B**) Cells differentiated with or without inhibitor treatments were harvested at indicated time points. Western blotting results show that O-GlcNAc levels in cells were efficiently suppressed or elevated by Ac5S (A) or TMG (T), respectively, as compared to control (Ct). (**C**) Representative images and quantification of multinucleated cells percentage and cilium length on cells differentiated with or without inhibitor treatments at indicated time points. Cells were harvested on days 0, 12, 25, 35, 45, and 55 and stained for IFT88, Arl13b, and NeuN (after day 25). Cilium lengths were measured on every ciliated cell on days 0, 12, and 25. Cilium lengths were measured on NeuN-positive cells after day 25. Scale bars for the image are 40 µm and for the magnified images are 10 µm. All data are mean ± SD from three independent experiments (>80 cells per experiment). *** *p* < 0.001; **** *p* < 0.0001; n.s., not significant.

**Figure 4 cells-12-01520-f004:**
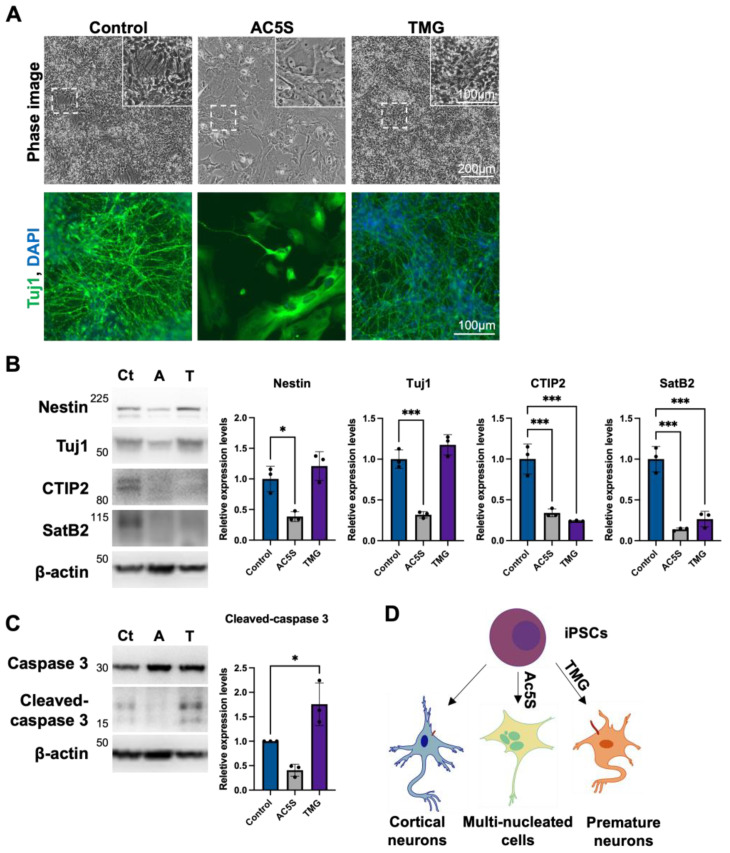
Altered OGN and subsequent cilia formation interferes with cortical neuronal differentiation. (**A**) Phase immunofluorescence images of Tuj1 staining. Cells differentiated with or without inhibitor treatment were observed on day 55. (**B**,**C**) Neuronal markers (**B**) and apoptosis marker (**C**) were detected by Western blotting on day 55. Ct: control, A: Ac5S, and T: TMG. (**D**) Schematics illustrating cells generated with or without inhibitor treatment possessed different cilium length, morphology, and identity. * *p* < 0.05; *** *p* < 0.001.

**Figure 5 cells-12-01520-f005:**
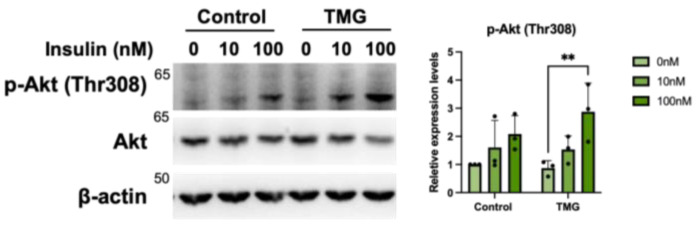
Increased primary cilium length leads to higher insulin sensitivity of TMG-treated neurons. Neurons differentiated under normal (control) and TMG-treatment conditions were serum-starved for 3 h prior to the addition of 10 or 100 nM insulin for 30 min. Lysates were analyzed by Western blotting and probed for pAkt (pT308), Akt and β-actin act as loading controls. (*n* = 3, ** *p* < 0.01).

## Data Availability

Data is contained within the article.

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
