# Peer review of "Regulation of Primary Cilium Length by O-GlcNAc during Neuronal Development in a Human Neuron Model"

_cells, 2023, doi:10.3390/cells12111520_

Round 1
Reviewer 1 Report
cells-2370616
Regulation of primary cilia length by O-GlcNAc during neuronal development in human neuron model by Tian and co-Authors.
The O-GlcNAcylation is a widespread modification of serine/threonine residues of nuclear, mitochondrial, and cytoplasmic proteins. Its level is regulated by two enzymes of reverse activity, O-GlcNAc transferase (OGT) and O-GlcNAcase (OGA). The O-GlcNAcylation is a nutrient sensor and changes of the level of O-GlcNAcylation are associated with pathogenesis of a number of human diseases.
The Authors investigate the correlation between primary cilia length and the level of O-GlcNAcylation in cortical neurons differentiated from human induced pluripotent stem cells (iPSC). Besides unravelling the negative correlation between the level of O-GlcNAcylation and primary cilia length (that agrees with previous studies on hTERT-RPE1 and IMCD3 cells as well as cilia of the diabetic mouse eyes, trachea, and OGT haploinsufficient mice), Authors analyzed also the effect of a long-term changes in O-GlcNAcylation level on neuron development and differentiation.
Overall, the manuscript is well written and data are clearly presented. The obtained results will be of interest not only to researchers investigating primary cilia assembly and length, and neurons development but also to the broader scientific community.
However, I have some minor suggestions to the Authors
The general remark is that immunofluorescence images and graphs, especially those showing cilia length, are way too small. There are plenty of empty space on the figures.
Other suggestions
1. It would interesting to check if the level of O-GlcNAcylation affects only primary cilia length or also the number of cells forming primary cilia (as was shown for other cell types),
2. Did Authors observed any effect on the intraflagellar transport (bulbs at the cilia tip suggesting accumulation of the ciliary proteins)?
3. Introduction, line 47, please add a name of the protein encoded by a Stump gene (B9 domain-containing protein 2)
4. Please explain abbreviation: HBP when it is first time used (for the readers outside the field)
5. Fig 1C and D – provided Figs’ descriptions are presented in reverse order
6. Fig 2A – please describe the stages of cortical neuron differentiation (morphological changes, markers?) in indicated time-points on a fig (or in fig description)
7. Fig 2C description; it is stated: “Cilia lengths were measured on NeuN-positive cells after day 25.” – but graph represent measurements of cilia length in all time-point. Please clearly state in which cells cilia were measured (all cells for samples 0-25 days, NeuN-positive cells for samples 35 days and later)
8. Fig 2: the significance at the level **P<0.01; ***P<0.001 are not shown on Fig 2 but are in Fig2 description
9. Fig 3B – in control sample the level of modification seems to be the highest after 25 days, not 35 days. Please explain. (more proteins loaded?).
10. Fig 3B Description. It seems that a word “respectively” should be directly after ” or TMG (T)”. Correct?
11. Fig 3C- IF images. Could Authors mark the boundary of the multinucleated cells by a white dotted line?
12. Fig 3. Description *P<0.05; **P<0.01; ***P<0.001 – significance at this level is not shown on the figure
13. Fig 4B-C – please quantify blots data (densitometry)
14. Lines 233-234:” TMG-treated cells were able to generate Tuj1-positive neurons though a reduced cell 233 number was noticed at later phases of differentiation.” – please provide graphs showing mentioned cell number differences. Is the difference statistically significant?
Reviewer 2 Report
The findings of this study provide association of O-GlcNAc levels with primary cilia morphology and cell fate determination of human iPSC-derived neurons.
Major points
To verify differentiation of human iPSCs into cortical neurons, authors should label primary cilia of differentiated cortical neurons using antibodies for adenylyl cyclase 3 and a subset of G protein-coupled receptors such as somatostatin receptor type 3 and serotonin receptor type 6, whose localization to neuronal primary cilia has been identified in the rodent cerebral cortex.
Authors claim that increased primary cilia length lead to high insulin sensitivity of TMG-treated neurons based on the results in Fig 5A. If so, authors should demonstrate that cilia elongation by pharmacological or genetic interventions that are unrelated to O-GlcNAcylation also increases insulin sensitivity, in order to exclude the possibility that TMG treatment induced high insulin sensitivity via a cilium-independent molecular pathway.
Minor points
In in Fig 1, a dramatic decrease in the O-GlcNAc level by Ac5S treatment and a mild decrease in the O-GlcNAc level by 2 mM glucose treatment elongated primary cilia to the same degree. Authors should explain this nonlinearity of the negative regulation of cilia length by O-GlcNAc levels.
Not only Fig 1A but also Fig 1C, Fig 2C and Fig 3C should include high magnification images that show precise cilia morphology. Scale bars should be added also in high magnification images.
Fig 2A and Fig 4A should include high magnification images that show precise cell morphology.
In Fig 3C, magnification for Day 0, 12 and 25 images and that for Day 35, 45 and 55 images seem to be not identical.
Authors note that “We determined primary cilia length from NeuN-positive cells by measuring the distance from the base to the tip as indicated by IFT88” in line 159-161. I cannot understand how Arl13b staining contributed to all experiments in this study.
The present findings obtained by differentiating human iPSCs into cortical neurons do not directly reflect implications of O-GlcNAc for neuronal development through its regulation of primary cilia in humans. Therefore, the limitations of this study should be noted regarding the gap between in vitro and in vivo roles of O-GlcNAc in neuronal development via primary cilia in the Discussion section.
Reviewer 3 Report
Cilia function in many physiological settings including neuronal differentiation. It has been shown that OGN level can disturb ciliogenesis while the physiological consequence has not been explored. This work demonstrate the level of OGN influence ciliary length as well as neuronal differentiation. The data is solid, the writing is clear and it is interesting to the ciliary and neuronal field.
If I understand correctly, the authors have not separated the effect of ciliary length from that of OGN level on signaling and/or neuronal differentiation, it is only correlation. In Figure 5, one could add colchicine to block ciliary elongation to see whether the signaling is triggered by changes in OGN level or by ciliary elongation. Another approach is to eliminate cilia by RNAi or CRISPR/CAS9.
Round 2
Reviewer 2 Report
Authors should describe how IFT88 staining and Arl13b staining complementarily served for cilia length quantification precisely in the manuscript.
Author Response
Thanks for the reviewer's comment. We have added the details of cilia length measurement in the manuscript. As following:
"We determined primary cilia length from NeuN-positive cells by measuring the distance from the base to the tip as indicated by IFT88, which excels at staining the entire cilia with the disadvantage of having a high background that makes it difficult to tell cilia apart. For this reason, we also used Arl13, which stains robust cilia axonemes, as a complement to locate and measure cilia length more specifically and accurately. "
Reviewer 3 Report
No more comments
Author Response
Thanks for the reviewer's comments.